# Simulation and Attribution Analysis of Spatial–Temporal Variation in Carbon Storage in the Northern Slope Economic Belt of Tianshan Mountains, China

Kun Zhang [1,2,3,4,5], Yu Wang [2,3,4,5], Ali Mamtimin [2,3,4,5,*], Yongqiang Liu [1], Lifang Zhang [1], Jiacheng Gao [2,3,4,5], Ailiyaer Aihaiti [2,3,4,5], Cong Wen [2,3,4,5], Meiqi Song [2,3,4,5], Fan Yang [2,3,4,5], Chenglong Zhou [2,3,4,5] and Wen Huo [2,3,4,5]

1. College of Geography and Remote Sensing Sciences, Xinjiang University, Urumqi 830046, China; zkun_@stu.xju.edu.cn (K.Z.); liuyq@xju.edu.cn (Y.L.); zhanglf@xju.edu.cn (L.Z.)
2. Institute of Desert Meteorology, China Meteorological Administration, Urumqi 830002, China; wangyu@idm.cn (Y.W.); gaojiach@idm.cn (J.G.); ailiyaer@idm.cn (A.A.); wencong@idm.cn (C.W.); songmq@idm.cn (M.S.); yangfan@idm.cn (F.Y.); zhoucl@idm.cn (C.Z.); huowenpet@idm.cn (W.H.)
3. National Observation and Research Station of Desert Meteorology, Taklimakan Desert of Xinjiang, Urumqi 830002, China
4. Taklimakan Desert Meteorology Field Experiment Station of China Meteorological Administration, Urumqi 830002, China
5. Xinjiang Key Laboratory of Desert Meteorology and Sandstorm, Urumqi 830002, China
* Correspondence: ali@idm.cn

**Abstract:** Intensive economic and human activities present challenges to the carbon storage capacity of terrestrial ecosystems, particularly in arid regions that are sensitive to climate change and ecologically fragile. Therefore, accurately estimating and simulating future changes in carbon stocks on the northern slope economic belt of Tianshan Mountains (NSEBTM) holds great significance for maintaining ecosystem stability, achieving high-quality development of the economic belt, and realizing the goal of "carbon neutrality" by 2050. This study examines the spatiotemporal evolution characteristics of the NSEBTM carbon stocks in arid regions from 1990 to 2050, utilizing a combination of multi-source data and integrating the Patch-generating Land use Simulation (PLUS) and Integrated Valuation of Ecosystem Services and Trade-offs (InVEST) models. Additionally, an attribution analysis of carbon stock changes is conducted by leveraging land use data. The findings demonstrate that (1) the NSEBTM predominantly consists of underutilized land, accounting for more than 60% of the total land area in the NSEBTM. Unused land, grassland, and water bodies exhibit a declining trend over time, while other forms of land use demonstrate an increasing trend. (2) Grassland serves as the primary reservoir for carbon storage in the NSEBTM, with grassland degradation being the leading cause of carbon loss amounting to 102.35 t over the past three decades. (3) Under the ecological conservation scenario for 2050 compared to the natural development scenario, there was a net increase in carbon storage by 12.34 t; however, under the economic development scenario compared to the natural development scenario, there was a decrease in carbon storage by 25.88 t. By quantitatively evaluating the land use change in the NSEBTM and its impact on carbon storage in the past and projected for the next 30 years, this paper provides scientific references and precise data support for the territorial and spatial decision making of the NSEBTM, thereby facilitating the achievement of "carbon neutrality" goals.

**Keywords:** scenario simulation; carbon storage; spatiotemporal evolution; InVEST model; northern slope economic belt of Tianshan mountains

## 1. Introduction

Globally, nations have set a goal of achieving "net zero carbon emissions" in 2021. Enhancing carbon storage in terrestrial ecosystems represents a pivotal approach to miti-

gating carbon dioxide emissions, thus constituting one of the foremost strategies [1]. The United Nations report on the 2030 Agenda for sustainable development underscores the imperative of safeguarding, rehabilitating, and sustainably harnessing ecosystems in order to advance sustainable development goals. Human activities and the process of economic development lead to modifications in land cover, resulting in consequential alterations to factors such as the land, climate, and environment. These changes subsequently impact global terrestrial carbon storage ecosystem processes [2]. Given the escalating concerns regarding land use, urban expansion, and climate change in the foreseeable future, they pose persistent threats to the sustainable development of terrestrial ecosystems and have garnered global attention [3,4]. Carbon stocks in terrestrial ecosystems are primarily affected by climate change and human-induced alterations in land use and cover, which is widely recognized [5]. On one hand, human activities and LUCC changes are substantial contributors to the rise in global temperatures through carbon emissions [6,7]. Controlling the global carbon cycle and climate change requires carbon storage, a key measure of the health of terrestrial ecosystems [8,9]. LUCC is the second major factor contributing to the significant rise in global carbon emissions, as multiple studies have demonstrated that they have a direct impact on terrestrial ecosystems' ability to store carbon [10,11]. On the other hand, human activities play a constructive role in carbon sequestration within arid regions through their influence on land use and vegetation cover dynamics. Purposeful human intervention and the implementation of ecological restoration policies have the potential to augment vegetation coverage and enhance greenery [12]. Hence, it is evident that land use change not only represents a prominent aspect of global surface environmental alterations but also serves as a focal point for investigating variations in carbon storage within terrestrial ecosystems. Consequently, accurately evaluating the impact of land use change on carbon storage and establishing harmonious, sustainable, and high-quality human–land relationships are imperative to accomplish the mission and objectives of achieving global "carbon neutrality" by 2050.

At present, there are three main approaches to studying carbon storage in terrestrial ecosystems: conducting field surveys, utilizing remote sensing technology, and employing model simulations. For instance, utilizing data from various carbon pools along with spatial–temporal visualization capabilities enables the effective calculation of carbon storage using models like InVEST [13]. The model is simple to operate, flexible in terms of parameters, and yields accurate results [14]. The InVEST model, when combined with GIS technology, has successfully addressed the limitations of traditional methods for estimating carbon storage. These limitations include extended sampling periods and intensive labor. In addition, the InVEST model offers the benefits of simplified parameter acquisition and visually presented results. As a result, it is extensively utilized in various applications [15–18]. Research has shown that the InVEST model, combined with geographic information mapping, was used to simulate carbon storage in the Poyang Lake basin [19]. In addition, some scholars have also used land use types in conjunction with the InVEST model to quantitatively assess the impact of LUCC conversion on ecosystem carbon storage [20]. Many researchers use models such as SD, FLUS, CA-Markov, ANN-CA, and logistic-CA to predict LUCC changes and have coupled these models with the InVEST model to reveal the spatial distribution characteristics of carbon storage under different future LUCC scenarios. The utilization of the Flus model and the InVEST model has yielded favorable validation outcomes in simulating carbon storage in the arid regions of northwest China [21]. Additionally, the CA-Markov model has demonstrated strong applicability in the Sariska Tiger Reserve in India. The utilization of the future land use prediction models can effectively explore the dynamic changes in carbon storage within the research area, thereby providing valuable support for informed decision making on land management by local departments and governments [22]. Among the abovementioned models used for future LUCC predictions, the PLUS model effectively elucidates the underlying causes of the diverse LUCC changes and accurately simulates spatial transformations in various small-scale regions. It facilitates the incorporation of prospective spatial policy elements

and enables a more scientifically rigorous simulation of the future LUCC alterations under distinct policy scenarios [23,24].

The north slope economic belt of Tianshan Mountains (NSEBTM) is located in a strip-shaped oasis at the foot of the northern slope of Tianshan Mountains, on the northwest border of China [25]. Due to its inland location, distance from the sea, and scarce precipitation, it belongs to China's arid region with fragile ecosystems and harsh climatic conditions [12]. The NSEBTM, endowed with abundant energy resources and strategically located, functions as a convergence zone propelled by the China–Central Asia–West Asia economic Corridor, the China–Russia–Mongolia economic Corridor, and the China–Pakistan economic Corridor. It assumes a pivotal role as an important strategic support point along the Silk Road economic belt and serves as a crucial bridge for China's western opening-up endeavors. In line with China's comprehensive opening-up strategy and modernization efforts, the NSEBTM holds significant strategic importance. In this scenario, the NSEBTM has emerged as the region with the highest level of urbanization and population density in Xinjiang [26]. However, in this resource-constrained and economically active arid region, the inherent contradiction between fragile ecosystems and high-quality economic and social development has persistently plagued the area. On one hand, there is an urgent imperative to achieve global "carbon neutrality" by 2050; on the other hand, compared to other regions in China, this particular region is situated within an arid zone where its environmental vulnerability exhibits heightened sensitivity to fluctuations in human–environment interactions [27]. As the pivotal region of the Belt and Road Initiative, the economic development and establishment of the NSEBTM as a free trade experimental zone will inevitably induce land use transformations, which in turn will have profound implications for carbon storage ecosystems [28]. Therefore, it is imperative to address these scientific inquiries. Presently, domestic and international research on the NSEBTM primarily concentrates on urbanization, urban spatial structure morphology, and landscape remote sensing monitoring, among other facets. However, further investigations are warranted to elucidate the underlying mechanisms governing the historical and future variations in carbon storage within the NSEBTM against the backdrop of achieving "carbon neutrality" by 2050. The mechanisms underlying the spatiotemporal variations in carbon storage within the NSEBTM region remain uncertain in the context of ecological engineering and urbanization.

In light of this, leveraging long-term time series remote sensing data products, this study employs the InVEST model to quantitatively analyze the spatiotemporal dynamics of carbon storage in the NSEBTM amidst land use change from 1990 to 2050. Additionally, the PLUS model is employed to construct multiple scenario simulations for natural, sustainable, and economic development by 2050. The objective is to investigate the future trends in carbon storage evolution and underlying impact mechanisms, thereby providing a decision-making foundation for well-organized urbanization and territorial spatial planning in the NSEBTM. Therefore, this study aimed to accomplish the following: (1) use LUCC data from 1990 to 2020 as well as LUCC data predicted using the PLUS model for disparate scenarios in 2050 and couple them with the InVEST model to analyze the spatiotemporal changes in LUCC and carbon storage in the NSEBTM from 1990 to 2050, (2) quantitatively assess the impact of LUCC changes on carbon storage in the NSEBTM by coupling LUCC data with the InVEST model, and (3) quantitatively assess the impact of ecological engineering construction and economic development strategies on LUCC change types, and their consequent effects on carbon storage dynamics in the NSEBTM. These studies will provide reliable data support for the NSEBTM LUCC management and decision making, filling the gap in research on the spatial and temporal changes in carbon storage under the background of LUCC change in the NSEBTM region, and providing a reference for achieving "carbon neutrality" goals in the NSEBTM and even China as a whole.

## 2. Materials and Methods

### 2.1. Study Area

The NSEBTM (40°52′ N∼47°14′ N, 79°53′ E∼96°23′ E) is situated in the Eurasian hinterland, deep within the inland region of northwest China (Figure 1). This region exhibits significant spatial heterogeneity in its natural conditions [25]. Encompassing an area of approximately 3.96 × 10⁵ km², it constitutes 23.8% of Xinjiang's total landmass. Being distant from the ocean, this area experiences a typical temperate continental climate influenced by the westerly circulation and moisture from the Arctic Ocean, resulting in limited precipitation but high evaporation rates accompanied by ample sunshine. The elevation in this region varies from −153 to 4814 m, with the Tianshan Mountain range exhibiting relatively higher altitudes while the surrounding basins display lower elevations [29]. The natural vegetation is sparsely distributed, predominantly comprising grasslands and desert flora. As a result of the "warm-wet transformation" of climate and the implementation of ecological restoration projects, Xinjiang's vegetation has gradually exhibited a discernible trend towards increased greening. Over the past decade, there has been an incremental rise in the Normalized Difference Vegetation Index (NDVI) at a rate of 0.005/10a [12]. According to previous research, the NSEBTM range mainly includes cities such as the Urumqi City, Karamay City, Turpan City, Hami City, Changji City, Fukang City, Kuitun City, Wusu City, Tacheng City, Bole City, Alashankou City, Shihezi City, Wujiaqu City, Huyanghe City, Shuanghe City, and Xinxing City [25]. In the past few years, with the progress of China's economy and the advancement of urbanization towards a phase of superior quality expansion, constructive human endeavors and initiatives for ecological restoration have had a notable impact on the field of ecological engineering in this particular area [12,30,31].

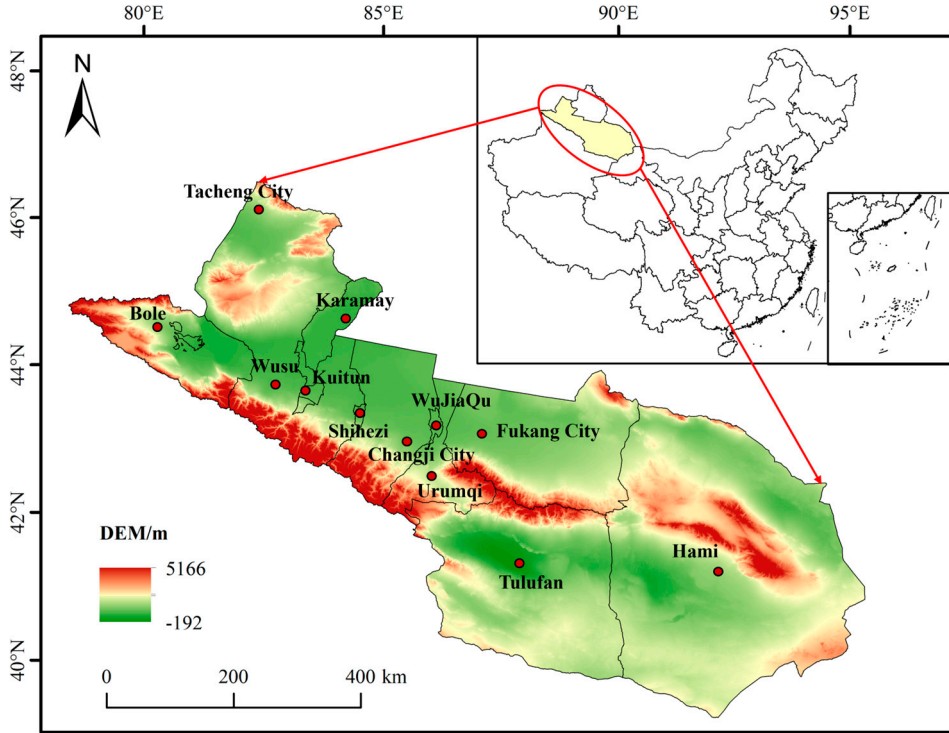

**Figure 1.** NSEBTM of Xinjiang, China. (Drawing review number: GS (2019) No. 1822. There is no modification to the base map, which is the same below).

### 2.2. Data and Methods

#### 2.2.1. Data Source

The data utilized in this study encompassed provincial administrative boundaries, land use and land cover change (LUCC), socioeconomic factors, climate factors, and terrain data (Table 1). Among them, the provincial administrative boundary data are in vector

format. The LUCC data were obtained from the first Landsat-derived annual China land cover dataset (CLCD); these datasets were derived from Landsat images available on Google Earth Engine [32]. The team first collected training samples through visual interpretation, combining China's land-use/cover datasets (CLUDs), satellite time series data, and samples extracted from Google Earth and Google Maps. They then constructed multiple temporal dimensions using Landsat images on GEE and applied a random forest classifier to obtain classification results. Finally, a spatiotemporal post-processing method was proposed to further improve the consistency of the spatiotemporal CLCD. The overall accuracy of CLCD reached 79.31%. To ensure consistency with other studies, we reclassified these LUCC types into six categories: farmland, forest, grassland, water, built-up land, and unused land. We selected LUCC data for 1990, 2000, 2010, and 2020 primarily to analyze changes in LUCC patterns as well as simulate carbon storage under different LUCC scenarios.

**Table 1.** Data types and sources.

| Type | Data | Resolution | Data Source |
|---|---|---|---|
| Provincial administrative boundaries | Research area boundaries | Vector data | Chinese Academy of Sciences Resource and Environmental Science and Data Center |
| LUCC | CLCD | 30 m | Wuhan University |
| Socioeconomic factors | GDP<br>Population | 1 km<br>1 km | Chinese Academy of Sciences Resource and Environmental Science and Data Center |
| | Distance from the city<br>Distance from the road<br>Distance from the water<br>Distance from the train station | Vector data<br>Vector data<br>Vector data<br>Vector data | OpenStreetMap |
| | Nighttime lighting data | 1 km | National Oceanic and Atmospheric Administration of the United States |
| Climate and environmental factors | Annual precipitation<br>Annual temperature<br>Soil type | 1 km<br>1 km<br>1 km | Chinese Academy of Sciences Resource and Environment Science and Data Center |
| | NDVI | 1 km | National Aeronautics and Space Administration |
| Topographic data | DEM | 30 m | Geospatial data cloud |
| | Slope<br>Aspect | 30 m<br>30 m | Based on ArcGIS |

The driving factor data include socioeconomic factors and terrain data. The socioeconomic factors include the Gross Domestic Product (GDP), population data, and distance data to cities, roads, water bodies, and stations. The terrain data include digital elevation model (DEM) data, slope, and aspect data obtained through surface analysis using DEM data in ArcGIS 10.8.1 software with a resolution of 30 m. The nighttime light data are used to characterize the urbanization process in the study area, while other driving factor data are mainly used for simulating future LUCC changes.

The climate and environmental factors include annual precipitation, temperature data, the number of soil types, and Normalized Difference Vegetation Index (NDVI) data. The NDVI data are used to represent the progress achieved in ecological restoration projects in the study area, while other climate and environmental factor data are used to adjust carbon density. The soil type data will also be involved in predicting the future LUCC data.

Considering the inconsistent spatial resolutions of the above data and the size of the study area, we standardized all the data to a spatial resolution of 500 m.

### 2.2.2. Human Activity Data Processing

Because of the different sensors used for the 2000 and 2020 nighttime light data, it was necessary to perform a data uniformity correction. First, the relative invariant target area method was used to perform relative correction on the 1992–2013 NSEBTM DMSP-

OLS images, combined with saturation and continuity correction after the radiometric calibration of the reference images. Simultaneously, noise processing and logarithmic transformation were used to correct the NPP-VIIRS images from 2012 to 2020. A significant correlation was found between the DN values of the DMSP-OLS images and the radiance values of the NPP-VIIRS images after index transformation. The BiDoseResp function model was then selected to perform a consistency correction on the DMSP-OLS and NPP-VIIRS images [33], ultimately obtaining a long-term time series of the nighttime light image datasets for the NSEBTM from 1997 to 2020.

Furthermore, because of the inconsistency in the DMSP-OLS data from different sensors for the same year, to fully utilize the image data obtained by different sensors, continuity correction was performed on the images after mutual and saturation correction. The continuity correction is divided into sensor and time-series continuity corrections. In this study, Equation (1) was used to perform sensor correction on the DMSP-OLS data, while Equation (2) was used to perform time-series continuity correction on both the DMSP-OLS andNPP-VIIRS data [34].

$$DN_{(i,j)} = \begin{cases} 0, & DN^a_{(i,j)} = 0 \quad \text{and} \quad DN^b_{(i,j)} = 0 \\ (DN^a_{(i,j)} + DN^b_{(i,j)})/2, & \text{other} \end{cases} \tag{1}$$

$DN_{(i,j)}$ is the *DN* value of the *j*th pixel in the corrected image in the *i*th year; $DH^a_{(i,j)}$ and $DH^b_{(i,j)}$ represents the *DN* value of the *j*th pixel in the images obtained from two different sensors before correction.

$$DN_{(i,j)} = \begin{cases} DN_{(i-1,j)}, & DN_{(i-1,j)} > DN_{(i,j)} \\ DN_{(i,j)}, & \text{other} \end{cases} \tag{2}$$

$DN_{(i,j)}$ and $DN_{(i-1,j)}$ represent the *DN* values of the *j*th pixel in the images corrected for saturation and multi-sensor continuity in the ith and *i*-1th year, respectively.

### 2.2.3. Carbon Storage Estimation Method

This study utilized the carbon storage sub-module of the InVEST model to calculate the carbon storage changes corresponding to LUCC changes over time for each remote sensing pixel. It simulates ecosystem carbon storage and carbon source/sink changes under different LUCC types and future land development scenarios [35]. To calculate ecosystem carbon storage, we used the carbon stock method to simulate carbon storage, which involves using carbon density data for aboveground, belowground, soil, and dead litter, multiplying the corresponding area by the carbon density data to calculate the storage data for each carbon pool, and then adding them together to obtain the total ecosystem carbon storage for a certain area [17]. The formula for calculation is presented below:

$$C_z = C_{z-above} + C_{z-below} + C_{z-soil} + C_{z-dead} \tag{3}$$

$$C_t = \sum_{i=1}^{n} C_z \times S_z \tag{4}$$

$C_z$ is the total carbon density ($t \cdot hm^{-2}$) of land-use type; $C_{z-above}$, $C_{z-below}$, $C_{z-soil}$, and $C_{z-dead}$ represent the aboveground carbon density, belowground carbon density, soil carbon density, and dead organic matter carbon density of LUCC type, respectively($t \cdot hm^{-2}$). $C_t$ is the total carbon stock of the ecosystem (*t*); $S_z$ is the area ($hm^2$) of LUCC type; *n* is the number of LUCC types (in this study, *n* is 6).

### 2.2.4. Selection and Correction of Carbon Density

The carbon density data for different LUCC types were obtained from the National Ecological Data Center's Resource Sharing Service Platform (http://www.nesdc.org.cn/, accessed on 10 February 2024), supplemented with relevant information from related research studies. Initially, we focused on selecting pertinent studies in the NSEBTM and

then expanded our search to include studies conducted in the arid and semi-arid regions of Xinjiang. Previous investigations have demonstrated that regional climate and soil type factors can significantly influence carbon density. Temperature exhibited a positive correlation with biomass, soil organic carbon density, and precipitation, while it displayed a negative correlation with them as well. To address this issue, we applied a correction method proposed by Alam et al. [36] to adjust the relationships between annual precipitation, biomass, and soil carbon density. The relationship between annual temperature and biomass carbon density was corrected using methods described by Giardina and Ryan [37] and Chen et al. [38]. Therefore, this approach was employed to rectify the carbon density data for various LUCC change categories within the study area, thereby obtaining localized carbon density data for the investigation region (Table 2). The specific approach is outlined as follows:

$$C_{SP} = 3.3968 \times MAP + 3996.1 \tag{5}$$

$$C_{BP} = 6.798 \times e^{0.0054 \times MAP} \tag{6}$$

$$C_{BT} = 28 \times MAT + 398 \tag{7}$$

$C_{SP}$ represents the soil carbon density corrected for annual precipitation; $C_{BP}$ represents the biomass carbon density corrected for annual precipitation; $C_{BT}$ represents the biomass carbon density corrected for annual temperature; $MAP$ represents the mean annual precipitation (mm); $MAT$ represents the mean annual temperature (°C).

$$K_{BP} = \frac{C'_{BP}}{C''_{BP}} K_{BT} = \frac{C'_{BT}}{C''_{BT}} \tag{8}$$

$$K_B = \frac{K_{BP}}{K_{BT}} K_s = \frac{C'_{SP}}{C''_{SP}} \tag{9}$$

$K_{BP}$ is the precipitation correction factor for biomass carbon density; $K_{BT}$ is the temperature correction factor for biomass carbon density; $K_B$ is the correction factor for biomass carbon density; $K_S$ represents the correction factor for soil carbon density; $C'$ and $C''$ are the carbon density data for the NSEBTM and the whole country, respectively.

**Table 2.** Carbon density variation among different LUCC types in NSEBTM.

| LUCC | $C_{above}$ | $C_{below}$ | $C_{soil}$ | $C_{dead}$ | References |
|---|---|---|---|---|---|
| Farmland | 4.18 | 3.38 | 80.22 | 0 | [20,21,39] |
| Forest | 44.51 | 3.37 | 137.12 | 0 | [20,21,39] |
| Grassland | 8.49 | 2.61 | 73.93 | 0 | [20,21,39] |
| Water | 0.92 | 0 | 0 | 0 | [20,21,39] |
| Built-up | 3.26 | 2.09 | 0 | 0 | [20,21,39] |
| Unused | 0.65 | 1.25 | 15.99 | 0 | [20,21,39] |

2.2.5. Scenario Simulation

When simulating the future LUCC map, three scenarios were set: natural growth, ecological protection, and economic development. (1) Natural growth scenario: this scenario is considered the baseline scenario, maintaining the historical trend of LUCC development and simulating future LUCC based on this trend. (2) Ecological protection scenario: in this scenario, ecological protection is strengthened, economic growth is slowed down, and the conversion of forests and grasslands into other land types is restricted. (3) Economic development scenario: This scenario focuses mainly on economic growth and neglects the protection of resources and the environment. Therefore, it restricts the conversion of built-up land to other land types, while increasing the probability of the other LUCC types being converted to built-up land.

The PLUS model was initially used to convert the LUCC data formats for 2010 and 2020. The LEAS module incorporated various driving factors to extract the expansion of

initial LUCC to final land use. To assess the development potential of the different LUCC types and investigate the relationship between these driving factors and LUCC expansion, we employed the random forest classification algorithm, which determined the contribution of each driving factor to LUCC expansion [40]; among them, the sampling rate of random forest was set to 0.01, and the running parameter was set to 5. The Markov chain method was used to predict future LUCC demand, and simulated patches were generated in the CARS module to obtain a simulated future LUCC map, among them, the default value within the domain range was set to 3, with 5 parallel threads, a decay coefficient of 0.9 for the decrement threshold, and a diffusion coefficient of 0.1. Based on adherence to the actual development situation in the research area and the transfer matrix law of land use area, three typical scenarios were set for each type of land's cost matrix (Table 3), where a value of "1" represented allowed conversion and a value of "0" represented otherwise [41–43]. The domain weights are shown in Table 4.

**Table 3.** Transition cost matrix for three land use scenarios.

| Land Use Scenarios | Land Use Types | Farmland | Forest | Grassland | Water | Built-up | Unused |
|---|---|---|---|---|---|---|---|
| Natural growth scenario | Farmland | 1 | 0 | 1 | 0 | 0 | 0 |
| | Forest | 1 | 1 | 0 | 0 | 0 | 0 |
| | Grassland | 1 | 1 | 1 | 1 | 1 | 1 |
| | Water | 1 | 0 | 1 | 1 | 1 | 1 |
| | Built-up | 0 | 0 | 0 | 1 | 1 | 0 |
| | Unused | 1 | 0 | 1 | 1 | 1 | 1 |
| Ecological protection scenario | Farmland | 1 | 1 | 1 | 1 | 0 | 1 |
| | Forest | 0 | 1 | 1 | 0 | 0 | 0 |
| | Grassland | 0 | 1 | 1 | 0 | 0 | 0 |
| | Water | 1 | 1 | 1 | 1 | 0 | 1 |
| | Built-up | 1 | 1 | 1 | 1 | 1 | 1 |
| | Unused | 1 | 1 | 1 | 1 | 0 | 1 |
| Economic development scenario | Farmland | 1 | 0 | 0 | 0 | 1 | 0 |
| | Forest | 0 | 1 | 0 | 0 | 1 | 0 |
| | Grassland | 0 | 0 | 1 | 0 | 1 | 0 |
| | Water | 0 | 0 | 0 | 1 | 1 | 1 |
| | Built-up | 0 | 0 | 0 | 0 | 1 | 0 |
| | Unused | 0 | 0 | 0 | 0 | 1 | 1 |

**Table 4.** Neighborhood weight settings.

| Land Use Type | Farmland | Forest | Grassland | Water | Built-up | Unused |
|---|---|---|---|---|---|---|
| Neighborhood Weight | 0.27 | 0.03 | 0.39 | 0.11 | 0.06 | 0.14 |

The evaluation of the PLUS model's simulation performance was conducted by measuring two metrics: overall accuracy (OA) and Kappa coefficient. The Kappa coefficient was computed according to the following formula:

$$\text{Kappa} = \frac{p_o - p_c}{p_p - p_c} \tag{10}$$

where Kappa is the simulation accuracy index, $P_o$ is the actual simulation accuracy, $P_c$ is the expected simulation accuracy under random conditions, and $Pp$ is the simulation accuracy under ideal conditions. Generally, when the Kappa value is greater than 0.75, the simulation accuracy is high; a value between 0.4 and 0.75 means that the simulation accuracy is moderate; and when it is less than 0.4, the simulation accuracy is poor. The Kappa coefficient of the simulation accuracy in this study was 0.86, the OA was 0.93, and the simulation results met the research requirements.

This research was conducted in three stages: (a) historical LUCC data from 2010 and 2020 were utilized along with the PLUS model to forecast LUCC data for 2050, with

separate predictions made for three scenarios; (b) carbon density data were adjusted based on temperature and precipitation information; and (c) the InVEST model was employed to estimate carbon storage by combining historical and projected LUCC data with carbon density data from 1990 to 2050. The methodological process is illustrated in Figure 2.

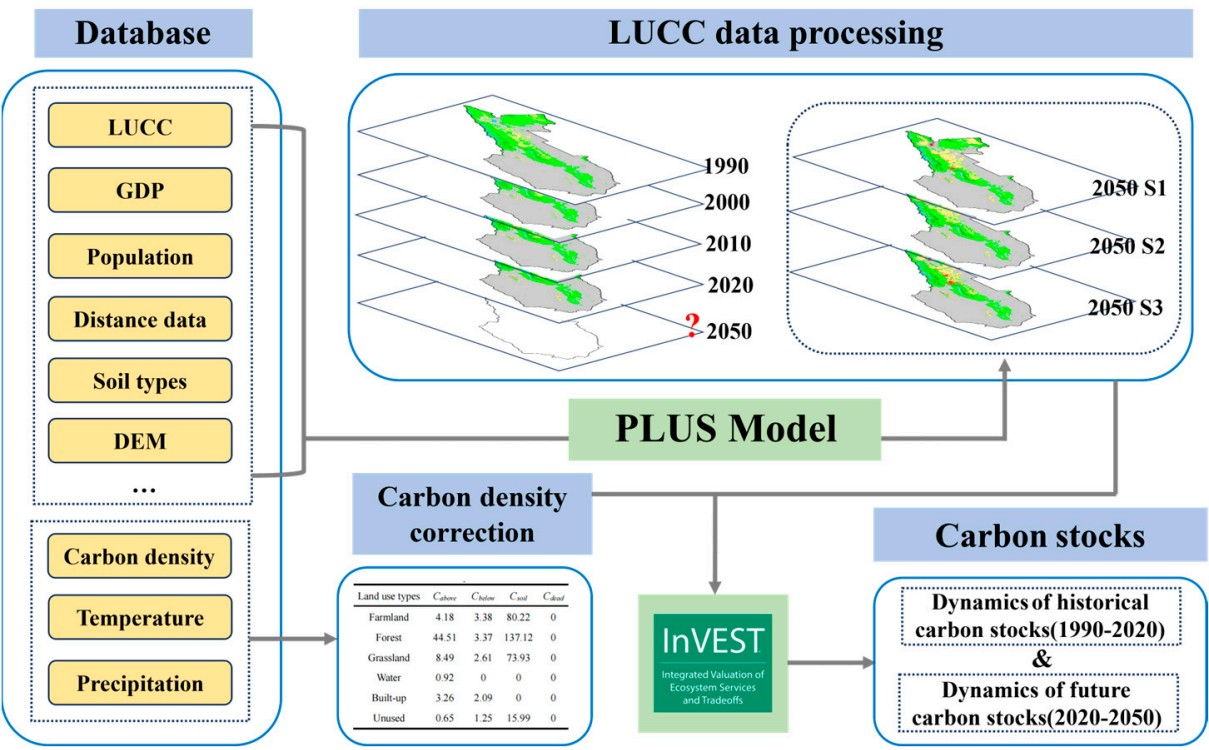

**Figure 2.** Methodology flowchart for this study.

## 3. Results

### 3.1. LUCC Dynamics in NSEBTM from 1990 to 2050

3.1.1. LUCC Dynamics from 1990 to 2020

In terms of spatial arrangement, owing to differences in terrain, landforms, and climatic conditions, the LUCC distribution characteristics of the NSEBTM show obvious spatial heterogeneity. As shown in Figure 3, arable land is mainly distributed in the southeastern part of the Tacheng region, Huyanghe City, and Karamay City. There are also scattered distributions in the central part of Hami City and the western part of the Turpan City. Forest land is relatively dispersed, mainly found in the sporadic areas of the Tianshan Mountains. Grassland is primarily distributed in the northwestern and southern parts of the Tacheng region, Shuanghe City, Urumqi City, as well as the central regions of the Hami City and Turpan City. Water bodies are mainly located in the western region of the Ili Kazakh Autonomous Prefecture. Construction land is predominantly distributed within urban clusters and their surrounding areas. Unused land is mostly found in large portions of the Hami City, Turpan City, and Changji Prefecture.

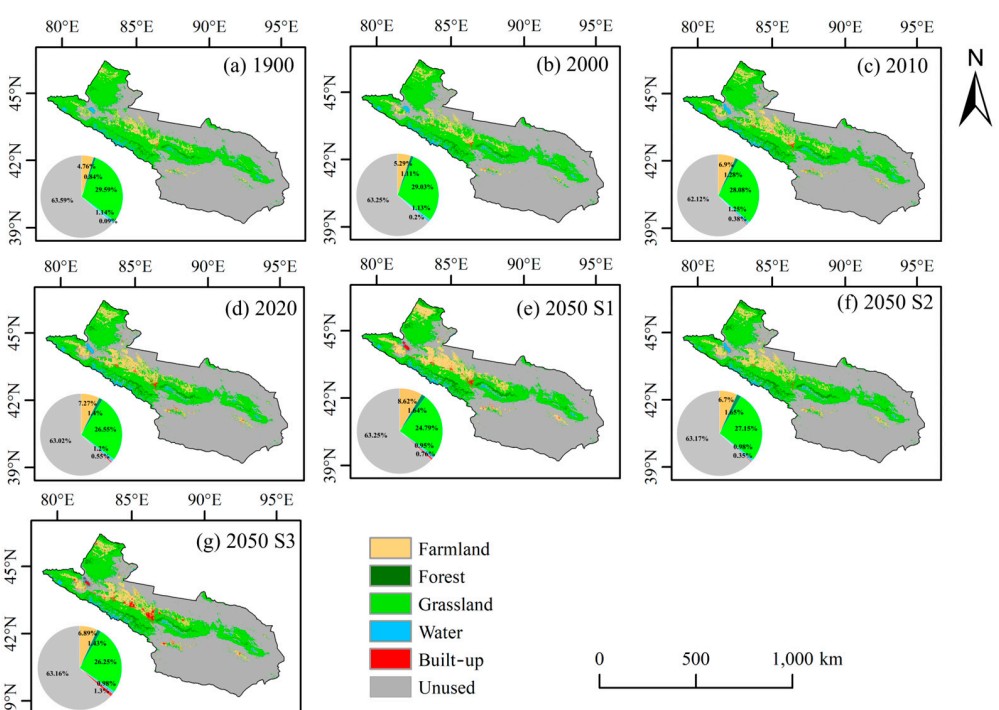

**Figure 3.** Spatial distribution map of LUCC in the NSEBTM from 1990 to 2050 and the proportion of LUCC types (S1 represents a natural development scenario; S2 represents an ecological protection scenario; S3 represents an economic development scenario).

The unused land area in the NSEBTM accounted for 63.59%, 63.25%, 62.12%, and 63.02% of the total area in 1990, 2000, 2010, and 2020, respectively, and was the most important LUCC type in the NSEBTM (Figure 3), with a much higher proportion than the national average (27.9%) [44]. Grassland and farmland were the next most important, accounting for 26.55–29.59% and 4.76–7.24%, respectively, with the proportion of farmland being much lower than that of the national average (14.3%) [44]. Water bodies, forest land, and built-up land had the smallest area, accounting for 1.13–1.25%, 0.84–1.40%, and 0.09–0.55% of the total area, respectively, with the proportion of built-up land being lower than the national average (4%) [44].

To clearly describe the changes in LUCC in the NSEBTM, we quantitatively expressed the conversion relationships between the different LUCC types using a Sankey diagram. The findings indicate that there have been diverse transformations in the LUCC categories in the NSEBTM during the last three decades (Figure 4). The area of unused land, which had the highest proportion, experienced an annual decrease. Specifically, 7394.17 km$^2$, 3744.89 km$^2$, 717.61 km$^2$, 9 km$^2$, and 718 km$^2$ were transformed into grassland, farmland, built-up land, forest land, and water bodies, respectively. Grassland also showed a decreasing trend every year, with 9558.73 km$^2$, 643.08 km$^2$, 2201.26 km$^2$, 342.45 km$^2$, and 9479.47 km$^2$ being converted to farmland, built-up land, forest land, water bodies, and unused land, respectively. However, it is worth noting that the area of farmland increased every year, with 2718.41 km$^2$, 463 km$^2$, 11 km$^2$, 98 km$^2$, and 100 km$^2$ being converted from grassland, built-up land, forest land, water bodies, and unused land, respectively. However, over the past 30 years, the areas converted from grassland, forest land, water bodies, and unused land to farmland were 9558.73 km$^2$, 8 km$^2$, 38 km$^2$, and 3744.89 km$^2$, and conversion into farmland far exceeded conversion from farmland. Forest land also showed an increasing trend, mainly being converted from grassland, farmland, water bodies, and unused land with areas of 2201.26 km$^2$, 11 km$^2$, 26 km$^2$, and 9 km$^2$, respectively. The main factor contributing to alterations in all types of LUCC was the unused land. The area of water bodies has remained stable over the past 30 years, with areas of 12,584.20 km$^2$ and 10,315.19 km$^2$ being converted from and to, respectively.

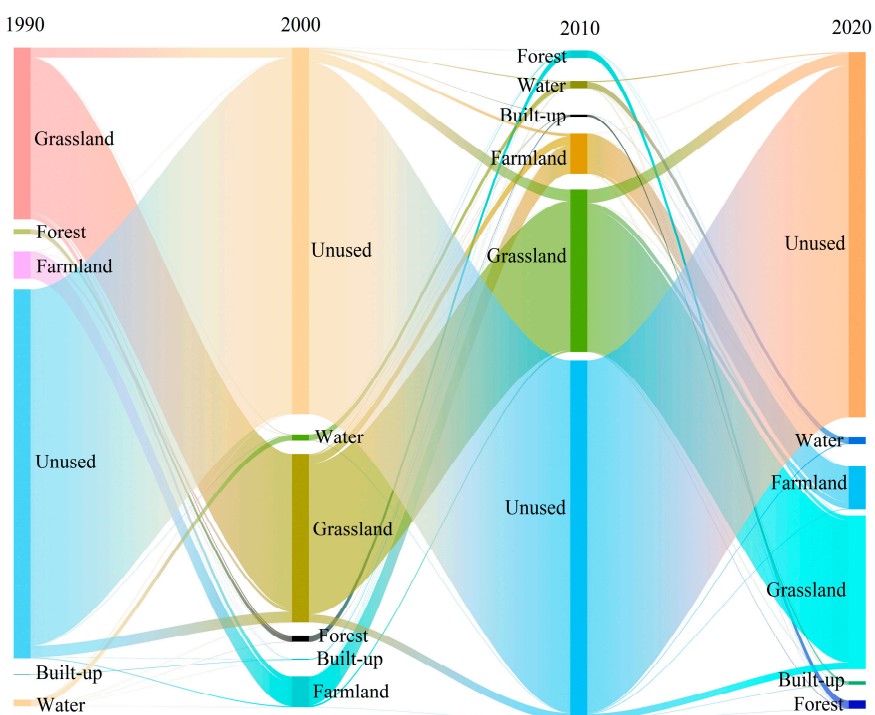

**Figure 4.** Dynamic changes in LUCC types in NSEBTM from 1990 to 2020.

### 3.1.2. LUCC Dynamics from 2020 to 2050

The validated PLUS model was employed to forecast the spatial distribution of LUCC change in 2050 under various scenarios, based on the LUCC data for the study area in 2020 (Figure 3e–g). Under the natural change scenario, the predominant LUCC type in 2050 is still unused land, accounting for 63.25% of the total area (Figure 3e), followed by grassland and farmland, accounting for 24.79% and 8.62%, respectively. Water bodies, forests, and built-up areas accounted for smaller proportions of the area (0.95%, 1.64%, and 0.76%, respectively). Grassland and water bodies continued to show decreasing trends, with cumulative areas decreasing by 6940.98 km² and 1010.38 km², respectively. Farmland, forests, built-up, and unused land are areas that showed cumulative increases in area of 5324.29 km², 942.03 km², 32.08 km², and 975.75 km², respectively. In terms of LUCC change direction (Figure 5a), all LUCC types showed trends of mutual transformation, with unused land being the main contributor to conversion to other LUCC types (8300.19 km², 1167.21 km², 378.43 km², 19 km², and 814.44 km² were converted to grassland, farmland, built-up areas, forests, and water bodies, respectively). Unused land remained the main contributor to LUCC change for the next 30 years.

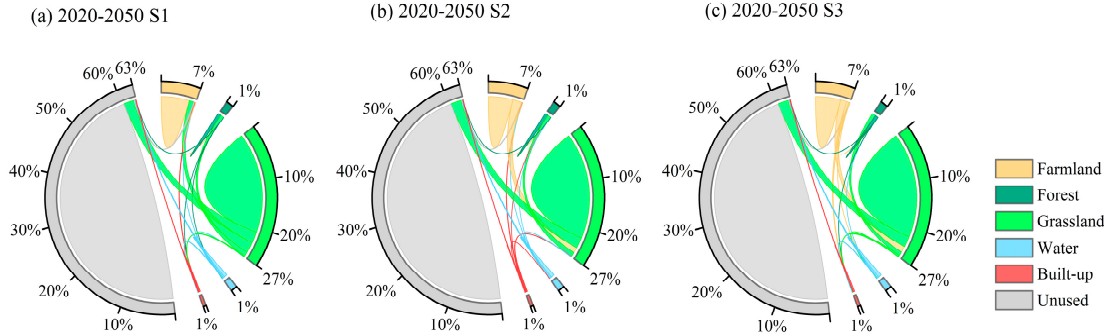

**Figure 5.** Dynamic changes in LUCC types in the NSEBTM in 2050 under different scenarios and the proportion of LUCC types. (S1 represents a natural development scenario; S2 represents an ecological protection scenario; S3 represents an economic development scenario).

Under the ecological protection scenario, the proportions of unused land, grassland, farmland, water bodies, forests, and built-up areas in the NSEBTM for 2050 were 63.17%, 27.15%, 6.70%, 0.98%, 1.65%, and 0.35%, respectively (Figure 3f). Compared to 2020, farmland, water bodies, and built-up land showed decreasing trends, with decreases of 2257.17 km$^2$, 874.93 km$^2$, and 792.40 km$^2$, respectively. Forests, grassland, and unused land showed increasing trends, with increases of 991.43 km$^2$, 2402.6 km$^2$, and 653.26 km$^2$, respectively. In terms of LUCC change direction (Figure 5b), all LUCC types showed similar trends of mutual transformation, with the increase in forest land mainly coming from the conversion of farmland to forests, and the increase in grassland mainly coming from the conversion of farmland, built-up land, and unused land. Overall, compared to the natural development scenario, the proportion of grassland increased from 24.79% to 27.15%, while the proportion of forests remained relatively stable, mainly because of the conversion of forests to grassland covering a larger area of 2669.87 km$^2$. There was relatively less conversion to other LUCC types.

In the context of economic development, it is projected that by 2050, the NSEBTM will have a distribution of unused land (63.16%), grassland (26.25%), farmland (6.89%), water bodies (0.98%), forest land (1.43%) and built-up land (1.30%) (Figure 3g). Compared with 2020, farmland, grassland, and water bodies showed significant decreases in area of 1509.66 km$^2$, 1151.62 km$^2$, and 865.42 km$^2$, respectively. The extent of built-up and unused land exhibited substantial increases, measuring 2938.08 km$^2$ and 611.74 km$^2$, respectively. However, there was a slight decline in the forested area. In terms of LUCC conversion (Figure 5c), all LUCC types showed a trend of mutual conversion, with conversion to built-up land being the most significant. Specifically, the areas of grassland, farmland, forest land, water bodies, and unused land that converted to built-up land were 1312.47 km$^2$, 1902.76 km$^2$, 8 km$^2$, 513.78 km$^2$, and 393.43 km$^2$, respectively. The primary driver of LUCC change in the economic development scenario was an increase in built-up land. In comparison to the natural development scenario, all other LUCC types experienced significant decreases, except for the proportion of built-up land which increased from 0.76% to 1.30%.

### 3.2. Dynamics of Carbon Storage in NSEBTM from 1990 to 2050

The InVEST model was employed to compute carbon storage in the NSEBTM from 1990 to 2050. However, the spatial distribution of carbon storage in the NSEBTM exhibited no significant changes (Figure 6). In general, high carbon reserves are mainly distributed in the central regions of the Tacheng City, Shuanghe City, Huyanghe City, Shihezi City, and Urumqi City. The highest values are found in scattered areas of the Tianshan Mountains. It is worth noting that the desert areas surrounding Hami City, Turpan City, and Changji Prefecture also store carbon reserves.

In order to enhance the clarity of illustrating the spatial changes in carbon storage in the NSEBTM, we conducted raster subtraction operations on the maps depicting carbon storage distribution during different time periods (Figure 7). According to the actual distribution of carbon storage, the areas were divided into carbon sink areas, balance areas, and carbon source areas, with values of 0 values and values close to 0 ($-500$ to 500 tons) being classified as balance areas [21]. From 1990 to 2000, 2000 to 2010, and 2010 to 2020, over 95% of the area did not show obvious changes in carbon storage, indicating that LUCC in most areas of the NSEBTM was minimally disturbed by human activities and did not undergo significant changes (Figure 7a–c). From 1990 to 2020, more than 3.60% of the total area showed an increase in carbon storage, indicating an enhancement in carbon sequestration capacity due to changes in LUCC, which led to an increase in carbon sink intensity. More than 3.18% of the area showed a decrease in carbon storage, indicating a weakening of the carbon sequestration capacity of the underlying surface, which turned into a carbon source area (Figure 7d). In general, the NSEBTM has maintained a relatively stable level of carbon storage over the past three decades, with approximately 95% of the region experiencing a balanced state between carbon storage gain and loss.

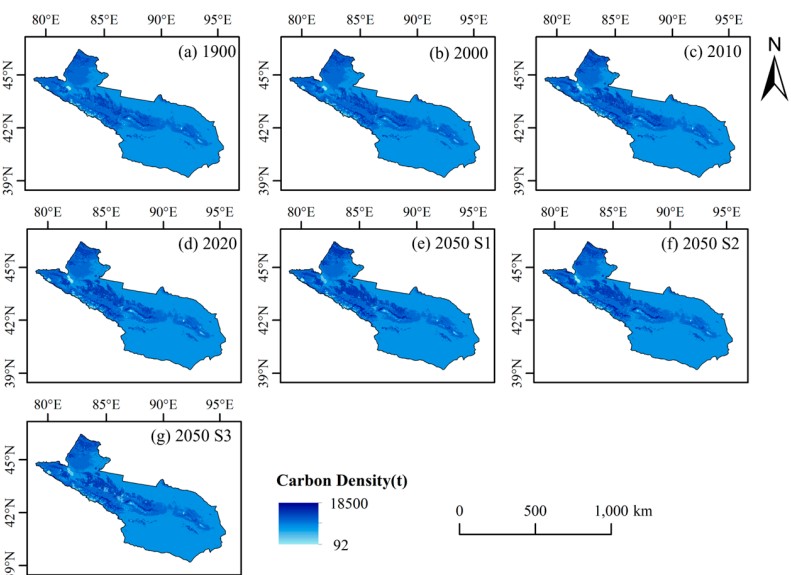

**Figure 6.** Spatial distribution of carbon storage in the NSEBTM from 1990 to 2050. (S1 represents a natural development scenario; S2 represents an ecological protection scenario; S3 represents an economic development scenario).

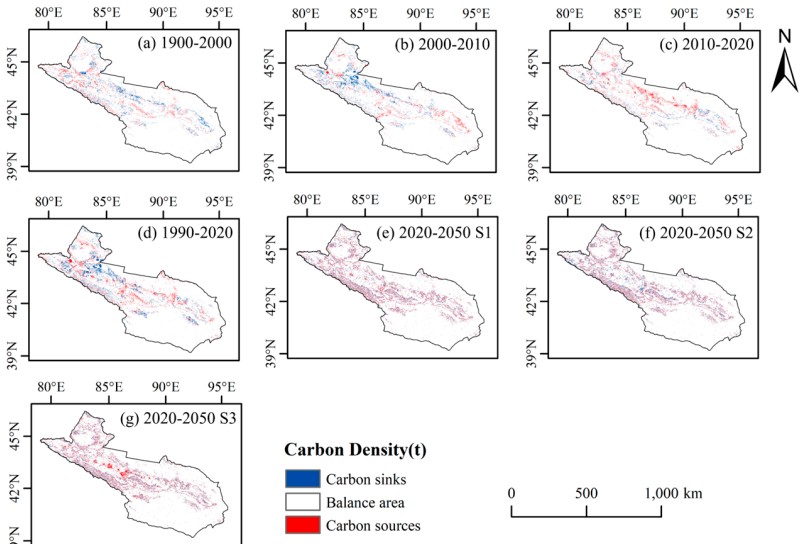

**Figure 7.** Spatial distribution changes in carbon storage in the NSEBTM from 1990 to 2050. (S1 represents a natural development scenario; S2 represents an ecological protection scenario; S3 represents an economic development scenario).

We also simulated carbon storage scenarios for the three development scenarios for 2050. Compared with 2020, in the natural development scenario, the areas with increased carbon storage exceeded 3.89%, the areas with decreased carbon storage exceeded 4.02%, and nearly 92.09% of the areas maintained a relatively stable carbon storage status (Figure 7e). In the ecological conservation scenario, areas with increased carbon storage exceeded 4.26%, areas with decreased carbon storage exceeded 3.60%, and over 92.14% of these areas maintained relatively stable carbon storage status (Figure 7f). Compared to the scenario of natural development, the ecological conservation scenario showed an increase in the area that absorbs carbon dioxide and a decrease in the area that emits carbon dioxide, while maintaining a relatively stable balance area. In the economic development scenario, areas with increased carbon storage exceeded 3.87%, areas with decreased carbon storage exceeded 5.20%, and nearly 91.74% of the areas were in a state of carbon balance (Figure 7g).

In the context of economic development, carbon sequestration decreases and the carbon source area expands compared to natural development, while the area of equilibrium remains relatively stable.

*3.3. Revisions in Carbon Storage Resulting from LUCC Change*

Based on the data presented in Table 5, the carbon storage contribution of various LUCC change types to the overall carbon storage can be ranked from highest to lowest as follows: grassland, unused land, farmland, forest land, water bodies, and built-up land. From 1990 to 2020, despite a declining trend in grassland areas (Figure 3) and a reduction of 102.35 t in carbon storage, it still constituted over 55% of the total carbon storage, thereby establishing itself as the predominant carbon sink. The area and carbon storage of unused land exhibited a declining trend, with a reduction of 4.05 t in carbon storage, constituting over 25% of the total carbon storage and positioning it as the second largest carbon sink within the region. The farmland area carbon storage exhibited a consistent annual increase, contributing to approximately 15% of the overall carbon storage. Over the past three decades, forest land witnessed a continuous growth in carbon storage, with an increment of 41.44 t. Although the forest land area constituted a relatively small proportion of the NSEBTM's total area, its share of carbon storage increased from 3.66% to 6.05%. Overall, the degradation of grasslands over the past three decades has been a significant contributing factor to carbon loss.

**Table 5.** Changes in carbon storage of LUCC types in NSEBTM from 1990 to 2050.

| LUCC | Carbon Storage/t | | | | | | |
|---|---|---|---|---|---|---|---|
| | **1990** | **2000** | **2010** | **2020** | **2050 S1** | **2050 S2** | **2050 S3** |
| Farmland | 165.33 | 183.65 | 239.69 | 252.75 | 299.49 | 232.93 | 239.50 |
| Forest | 61.20 | 81.20 | 93.55 | 102.64 | 120.07 | 120.95 | 104.47 |
| Grassland | 995.93 | 976.99 | 945.24 | 893.58 | 834.65 | 914.10 | 883.87 |
| Water | 0.41 | 0.41 | 0.46 | 0.44 | 0.35 | 0.36 | 0.36 |
| Built-up | 0.18 | 0.43 | 0.80 | 1.17 | 1.62 | 0.75 | 2.75 |
| Unused | 450.40 | 447.95 | 439.94 | 446.35 | 448.09 | 447.52 | 447.44 |

We also calculated the carbon storage of each LUCC type in 2050 under the three development scenarios. Under the scenario of natural development, grassland and water bodies experienced a decrease in carbon storage by 58.93 t and 0.09 t, respectively, compared to 2020. Meanwhile, unused land, farmland, forest land, and built-up land saw an increase in carbon storage by 1.74 t, 46.74 t, 17.43 t, and 0.45 t, respectively. Under the ecological protection scenario, grassland and forest land exhibited an increase in carbon storage by 79.45 t and 18.31 t, respectively, compared to the levels observed in 2020. Conversely, farmland, built-up land, and water bodies experienced a decrease in carbon storage by 19.82 t, 0.42 t, and 0.08 t correspondingly. In the context of economic development, a significant increase in carbon storage was observed on developed land compared to the levels recorded in 2020, with an increment of 1.58 t, while the changes in the carbon storage of the other LUCC types were relatively inconspicuous. Overall, in the three development scenarios for 2050, grassland remained the most important carbon sink in the NSEBTM, and the increase or decrease in the grassland area mainly affected the carbon balance in the NSEBTM.

Changes in LUCC type significantly affected changes in carbon storage. Based on the changes in LUCC transformation as well as differences in soil and vegetation density from 1990 to 2050, the impact of LUCC changes on carbon storage in the NSEBTM was calculated. As shown in Figure 8, the changes in carbon storage from 1990 to 2020 were mainly caused by the conversion between grassland and unused land. From 1990 to 2020, the conversion from grassland resulted in a decrease of $7.17 \times 10^7$ t in the region's carbon storage, and there was an increase of $2.20 \times 10^7$ t in carbon storage from conversion to forest land, a decrease of $2.88 \times 10^6$ t from conversion to water bodies, a decrease of $5.12 \times 10^6$ t from conversion to built-up land, and a decrease of $6.37 \times 10^7$ t from conversion to unused land. This decrease was much greater than the increase, indicating that conversion from grassland was not conducive to increasing regional carbon storage. The conversion of a substantial amount of

unused land into farmland and grassland resulted in respective increases of $2.62 \times 10^7$ t and $4.97 \times 10^7$ t in carbon storage. A minor conversion between unused land and water bodies resulted in a basic balance between the increase or decrease in carbon storage. Similarly, the bidirectional conversion between unused land and grassland resulted in a "balance between gains and losses" in the regional carbon storage. In the natural progression from 2020 to 2050, there was a net decrease of $6.15 \times 10^7$ t in carbon storage due to the transformation of grassland into unused land. Conversely, the conversion of this unused land into forest resulted in an increase of $3.59 \times 10^7$ t in carbon storage. Additionally, there was a significant boost of $5.57 \times 10^7$ t in carbon storage when converting unused land back into grassland, and a further increase of $8.16 \times 10^6$ t when transforming it into farmland. In the context of ecological conservation, between 2020 and 2050, the transition from grassland to forest land resulted in a carbon storage increase of $3.65 \times 10^7$ t. Similarly, converting water bodies to grassland and unused land led to respective increments of $3.73 \times 10^6$ t and $2.02 \times 10^6$ t in carbon storage. Furthermore, transforming built-up land into farmland and grassland contributed to an additional carbon storage of $4.29 \times 10^6$ t and $5.66 \times 10^6$ t, respectively. Notably, there were significant increases in carbon storage by converting unused land into farmland ($6.99 \times 10^6$ t) and grassland ($5.85 \times 10^7$ t). In terms of carbon storage decrease, the conversion from farmland to unused land resulted in a reduction of $6.53 \times 10^6$ t in carbon storage. The conversion from forest land to grassland led to a substantial decline of $2.67 \times 10^7$ t in carbon storage. Similarly, the transformation from grassland to water bodies and unused land caused significant decreases of $1.51 \times 10^6$ t and $5.91 \times 10^7$ t in carbon storage, respectively. Overall, in the ecological protection scenario, the increase in carbon storage resulting from land type conversion was much greater than the decrease, the main manifestation was the conversion of other LUCC types to grassland, resulting in a stronger regional carbon storage capacity. Under the economic development scenario, the main conversion was between grassland and unused land, and the increases and decreases in carbon storage were generally balanced. The conversion of farmland and grassland into built-up land resulted in reductions of $1.57 \times 10^7$ t and $1.05 \times 10^7$ t in carbon storage, respectively. Overall, under the economic development scenario, the decrease in carbon storage exceeded the increase, mainly due to the conversion of other LUCC types to built-up land, leading to a net loss of approximately $1.79 \times 10^7$ t.

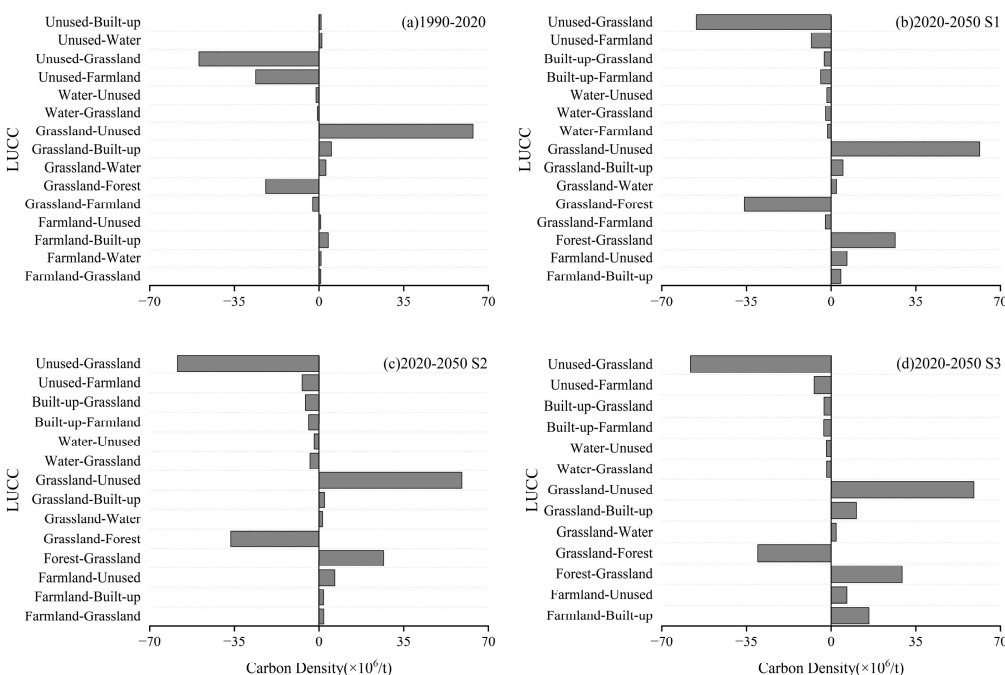

**Figure 8.** Changes in carbon stock caused by LUCC type changes in NSEBTM from 1990 to 2050.

## 4. Discussion

### 4.1. Implications of Human Actions on Changes in LUCC

The NSEBTM, located on the western frontier of China, has a large regional area and is one of the most sparsely populated areas in China's geographical space [12]. The NSEBTM is deep inland and far from the sea with a closed terrain, which makes it difficult for marine moisture to reach the region. It is controlled by continental air masses throughout the year, with scarce precipitation and an arid climate [45], which pose significant challenges to vegetation growth in the region. The main LUCC types in the NSEBTM are unused land, grassland, and farmland. Although the forest land area is relatively sparse, it is particularly important in LUCC planning, especially in the ecologically fragile areas of the NSEBTM. In 1979, China planned to implement the "Three-North Shelterbelt Project", which has a planned duration of 73 years, is divided into eight phases, and aims to establish large-scale artificial forestry. As an important part of northwest China, the NSEBTM has made substantial progress in ecological engineering construction under the Three-North Shelterbelt Policy [46]. By 2020, the fifth phase of the "Three-North Shelterbelt Project" in China was almost completed, and the forest area in the NSEBTM increased from 3307.17 km$^2$ in 1990 to 5546.42 km$^2$ in 2020 (Figure 3). NDVI data can characterize vegetation coverage; therefore, we established a spatial correlation between the NSEBTM NDVI data and the carbon storage data simulated using the InVEST model (Figure 9a). The graph reveals a distinct correlation between NDVI and carbon storage, particularly in the Huyanghe City, Karamay City, and Yili Prefecture. Taking the central region of the Yili Prefecture as an illustrative example from 2000 to 2020, there has been a notable augmentation in vegetation coverage within this area. Consequently, the expansion of vegetation coverage has led to a concurrent increase in carbon storage (Figure 9b,c).

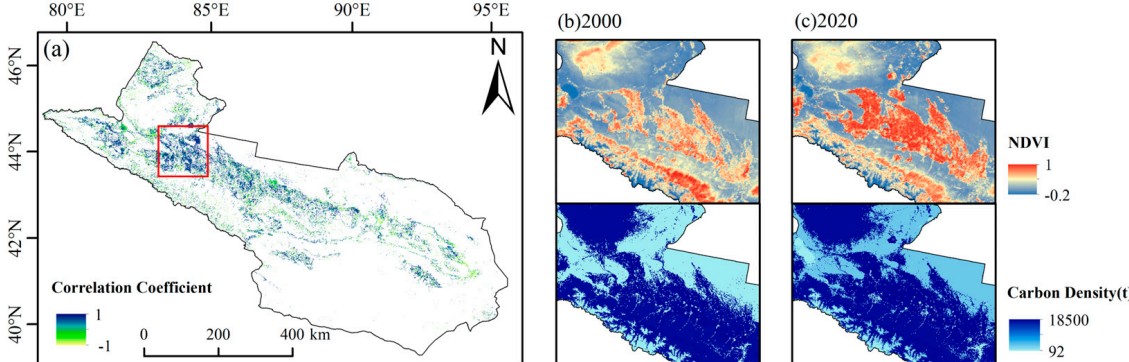

**Figure 9.** Changes in NDVI and corresponding variations in carbon storage. (**a**) Correlation between carbon stock and NDVI; (**b**) spatial distribution of NDVI and carbon stock in local areas in 2000; (**c**) spatial distribution of NDVI and carbon stock in local areas in 2020.

From 1990 to 2020, the farmland area in the NSEBTM increased from 18,832.5 km$^2$ in 1990 to 28,791.71 km$^2$. In 2021, the Xinjiang Uygur Autonomous Region issued the "14th Five-Year Plan for the Protection and Development of Land Resources in Xinjiang Uygur Autonomous Region", which mentioned the management policy for farmland: strengthening the protection of farmland, focusing on curbing the "non-agriculturalization" and preventing the "non-foodization" of farmland, and strictly observing the red line and bottom line of farmland. The overall LUCC plan issued by the Xinjiang Uygur Autonomous Region (2006–2020) also emphasized the effective protection of farmland, which plays a very important role as the third largest carbon sink in the NSEBTM. Over the past 30 years, the grassland area in the NSEBTM has shown a decreasing trend, and under our predicted natural development scenario for 2050, the grassland area will continue to decrease without human intervention. Conversely, under the ecological protection scenario, which adheres to the policy of protecting forests and grasslands, the grassland area will continue to increase. Overall, LUCC types in the NSEBTM are strongly influenced by

human activities, which means that in the future, it will be necessary to reasonably restrict human activities to protect the ecological land in the NSEBTM.

### 4.2. Impact of Urbanization on Carbon Storage in NSEBTM

The findings of the research indicate that the process of rapid urbanization and economic growth has significantly expedited land degradation [47]. The expansion of urbanized land area signifies the process of urbanization, and it is crucial not to underestimate the conversion from non-built-up land to built-up land, as it results in a reduction in carbon storage. Therefore, we used nighttime light remote sensing data to characterize the changes in built-up land types in the NSEBTM. According to the statistics of the NSEBTM's illuminated area (Figure 10(b-3,b-4,c-3,c-4)), the illuminated area expanded four-fold from 2000 to 2020, increasing from 2.42% in 2000 to 7.20% in 2020. The spatial distribution of nighttime light data shows that built-up land has expanded to surrounding areas based on the original area, unlike the increase in light intensity in coastal areas. Urbanization in the inland areas of the NSEBTM is manifested by the expansion of nighttime lighting areas. The increase in nighttime lighting areas contributed to a reduction in carbon storage. With the development of urbanization in the NSEBTM, three central cities will be built, driving the joint development of the Urumqi metropolitan area, northern Xinjiang urban belt, and southern Xinjiang urban cluster. It is evident that, under the guidance of urbanization policies in Xinjiang, from 2000 to 2020, there has been an obvious increase in built-up land in the NSEBTM (Figure 10b,c), corresponding to a decrease in carbon storage.

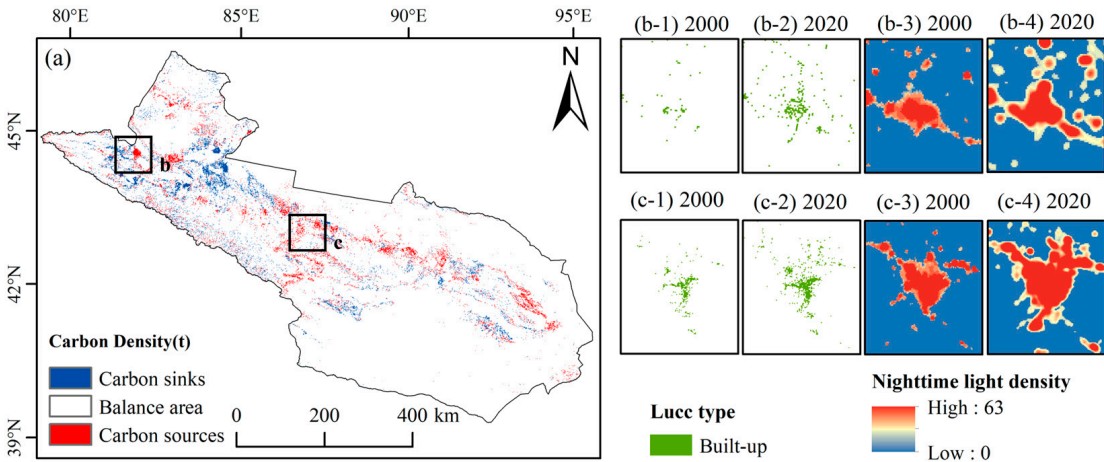

**Figure 10.** The relationship between built-up land change and carbon storage. (**a**) Spatial changes in carbon storage in the NSEBTM from 2000 to 2020, The black boxes in Figure (**a**) correspond to Figure (**b**) and Figure (**c**), respectively; (**b,c**) the distribution map of urban built-up land, and the spatial distribution map of light remote sensing data.

Research has shown that the level of urbanization in China has increased obviously from 2000 to 2019, and industrialization has accelerated obviously. Urban development requires more built-up land, and the continuous expansion of built-up land areas means that more energy and industrial production activities will occur, leading to a loss of carbon storage [48]. Numerous studies have quantitatively demonstrated that an increase in urban built-up land leads to a loss of carbon storage, such as the 8.64 Tg loss of carbon storage in the Chang-Zhu-Tan urban agglomeration due to the urban expansion from 1995 to 2009 [49]. In addition, numerous studies have shown that an increase in urban land will lead to a large amount of artificial greening and landscaping activities, which, although helpful in increasing urban green coverage, will also have adverse effects on soil quality and soil carbon storage capacity [50,51]. Furthermore, an increase in urban land will lead to an increase in traffic and industrial pollution, which will adversely affect soil and vegetation, further reducing carbon storage [52].

### 4.3. Implications of Different Scenarios of Carbon Storage Results for Future Planning

Our findings demonstrate that, under the ecological protection scenario for 2050, there was a significant increase in carbon storage by 12.34 t compared to the natural development scenario (Table 5). Notably, grasslands emerged as the primary carbon reservoirs in the NSEBTM, contributing over 55% of the total carbon storage (Figure 6). The degradation of grassland in the NSEBTM has been identified as the primary factor contributing to the loss of carbon storage, resulting in a significant decline of 102.35 t over the past three decades. The NSEBTM has abundant grassland resources and a variety of grassland types and species [53]; therefore, in future planning, it is necessary to strengthen the protection and rational utilization of grassland and steppe resources to increase carbon storage. Unused land was the main LUCC type in the NSEBTM, with deserts being the predominant LUCC type. Deserts are an important part of terrestrial ecosystems, and their contribution to carbon storage cannot be ignored. Gulnur et al. [54] explained that the Gurbantunggut Desert is a carbon sink during the vegetation growing season. Similarly, the Taklimakan Desert in southern Xinjiang is also an important carbon sink, sequestering $148.85 \times 10^4$ tons a$^{-1}$ of carbon annually [55]. Our study confirms this conclusion, with unused land accounting for over 25% of the total carbon storage in the NSEBTM, making it the second largest carbon reservoir among the NSEBTM's land types. Therefore, given the NSEBTM's unique geographical location and climatic conditions, it is important to develop desert ecological projects that effectively increase land carbon storage. In future plans, investments in desert ecological projects should be increased to enhance carbon storage. Additionally, it is worth noting that forests and farmland are important sources of carbon storage in terrestrial ecosystems. Although the forests and farmlands in the NSEBTM occupy relatively small areas and are dispersed, their contribution to carbon storage should not be overlooked. Studies have shown that extensive logging and grazing activities reduce the carbon stored in shrubs, trees, roots, litter, and dead plants [56]. In related studies on farmland carbon storage, researchers have shown through 40 years of field cultivation data that fallow farming can increase soil organic carbon storage in the Yellow River Delta farmland [57]. Therefore, it is necessary to plan the use of forest and farmland resources in a rational manner to increase carbon storage.

Our study also shows that, compared to natural development scenarios, economic development scenarios resulted in a loss of 25.88 t of carbon storage (Table 5). Urbanization leads to an expansion of built-up land, resulting in a reduction in vegetation cover and subsequently diminishing carbon storage capacity. Furthermore, urbanization induces alterations in land use patterns, such as the conversion of farmland or forests into developed areas, leading to a decline in carbon storage. In the future process of economic development, it will be necessary to strengthen urban greening and ecological construction, protect and restore vegetation cover around cities, and increase urban green space and forest cover to increase urban carbon storage. In addition, it is necessary to strengthen urban planning and management, rationally use land resources, protect farmland and forest resources, and reduce the impact of land development on carbon storage.

In conclusion, based on the impact of LUCC changes on carbon storage in different scenarios, our findings provide a scientific basis and reference for LUCC planning and can help the government and relevant departments better formulate future LUCC policies and plans, promote ecological protection and sustainable development, and achieve a virtuous cycle of ecological environment and economic development.

### 4.4. Potential Applications and Limitations

The novelty of this study lies in the comprehensive analysis of the spatiotemporal variations in the NSEBTM carbon storage under past and future 30-year land use change scenarios using the PLUS model and InVEST model. By combining NDVI data and nighttime light remote sensing data, it reflects the changes in the NSEBTM carbon storage under the background of ecological engineering and urbanization. The advantage of the PLUS model over other land use prediction models is its application of a new analytical strategy that

includes a new multi-seed growth mechanism coupled with multi-objective optimization algorithms, which can better support planning policies for sustainable development. The high accuracy of the future LUCC simulation results conforms to the development patterns under different scenarios for the NSEBTM, thus providing an approach to simulate the regional LUCC and carbon storage, serving as an example and reference for carbon storage research in other regions, and promoting the sustainable development of the NSEBTM.

This study also has certain limitations. The future LUCC patterns of the NSEBTM will change due to factors such as climate change, natural disasters, and policies. As the "warm-wetting" trend intensifies in northwest China [58], the LUCC patterns will also undergo changes, which will increase the uncertainty of the LUCC predictions. Additionally, carbon density will also change over time. Future research will establish models that will capture the relationship between carbon density and time to predict data that will align with future periods.

## 5. Conclusions

We conducted an analysis using the PLUS and InVEST models to examine how LUCC and carbon storage in the NSEBTM have changed over the past three decades, as well as projected changes for the next 30 years. Additionally, we performed a quantitative assessment to determine the factors contributing to variations in carbon storage under different scenarios. Our findings indicate that between 1990 and 2020, unused land was the primary type of LUCC change observed in the NSEBTM, followed by grassland. By 2050, different development scenarios will induce alterations in land use; nevertheless, unused land will remain the dominant category. Over the past three decades, grasslands have served as a crucial carbon sink in the NSEBTM with substantial carbon sequestration capacity compared to other types of land such as unused land. Notably, the degradation of grasslands has been identified as a key driver behind the declining carbon storage levels within this region. Under diverse development scenarios for 2050, ecological conservation initiatives are expected to contribute towards energy savings along with reduced emissions while simultaneously enhancing terrestrial carbon stocks; however, urban expansion particularly under economic development scenarios may result in diminished carbon storage capacities. Therefore, it is imperative to strategically plan future LUCC considering economic development objectives alongside tailored management strategies for distinct LUCC types aiming at mitigating the potential losses of carbon storage within the NSEBTM.

**Author Contributions:** Conceptualization, K.Z. and A.M.; methodology, K.Z. and Y.L.; software, Y.W.; validation, C.W. and M.S.; formal analysis, F.Y., C.Z. and W.H.; data curation, J.G. and A.A.; resources, A.M.; writing—original draft preparation, K.Z.; writing—review and editing, K.Z., A.M. and Y.L.; visualization, K.Z. and L.Z. All authors have read and agreed to the published version of the manuscript.

**Funding:** This research was funded by the National Natural Science Foundation of China, grant number: 42305132 and 42375084, the Scientific and Technological Innovation Team (Tien Shan Innovation Team) project, grant number: 2022TSYCTD0007, the Third Xinjiang Scientific Expedition, grant number: 2022xjkk0903, the Special Funds for Basic Scientific Research Business Expenses of Central-level Public Welfare Scientific Research Institutes, grant number: IDM2021005, the Special Project for the Construction of Innovation Environment in the Autonomous Region, grant number: PT2203, and the Graduate Education Innovation Program of the Autonomous Region, grant number: XJ2023G032.

**Data Availability Statement:** The data are available on request.

**Conflicts of Interest:** The authors declare no conflicts of interest.

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
