# Peer review of "Simulation and Attribution Analysis of Spatial–Temporal Variation in Carbon Storage in the Northern Slope Economic Belt of Tianshan Mountains, China"

_land, doi:10.3390/land13050608_

Round 1

Reviewer 1 Report

Comments and Suggestions for Authors

I appreciate the opportunity to review this manuscript entitled "Simulation and Attribution Analysis of Spatial-Temporal Variation in Carbon Storage in the Northern Slope Economic Belt of Tianshan Mountains, China." This paper is about carbon dynamics in arid ecosystems, particularly in the context of land use changes and their implications for carbon storage. I think it's very interesting. Below are some specific comments:

(1) The introduction section  could more sharply delineate the research gaps your study aims to fill.

(2) Please provide more detailed information on the parameterization of the PLUS and InVEST models would enhance the replicability of your study. For example, detailing the assumptions made in the ecological conservation and economic development scenarios would add depth to your analysis.

(3) The results are compelling, especially the nuanced understanding of how different land use scenarios impact carbon storage. Expanding on the implications of these findings for land management policies in the region could make the study even more impactful.

(4) The discussion section could be strengthened by drawing more direct connections between your findings and broader ecological theories or models. Additionally, discussing the limitations of your study, such as the potential impacts of climate change on the model's accuracy, would provide a more balanced view.

(5) Ensure that all references are current and relevant, with a particular focus on recent studies that have employed similar modeling approaches in arid or semi-arid ecosystems.

Reviewer 2 Report

Comments and Suggestions for Authors They studied the carbon storage changes in different kind soils. It revealed that from 1990 to 2020, unused 668 land was the predominant LUCC type in NSEBTM, followed by grassland. By 2050, different development scenarios will induce alterations in land use; nevertheless, unused land will remain as the dominant category. Over the past three decades, grasslands have 671 served as a crucial carbon sink in NSEBTM with substantial carbon sequestration capacity 672 compared to other types of lands such as unused land.

I think the idea that old data also can help to estimate carbon storage is original. Furthermore, the visualization also very impressive. I think it will be better, if the authors put the measurement methodology of the different data. E.G. how measured the Wuhan University. Please change the figures, because these are hard to read.
